# Elevated *MMP9* Expression—A Potential In Vitro Biomarker for COMPopathies

**DOI:** 10.3390/ijms262412070

**Published:** 2025-12-15

**Authors:** Helen F. Dietmar, Ella P. Dennis, Francesca M. Johnson de Sousa Brito, Louise N. Reynard, David A. Young, Michael D. Briggs

**Affiliations:** Biosciences Institute, Faculty of Medical Sciences, Newcastle University, Newcastle upon Tyne NE1 3BZ, UK; helen_dietmar@hotmail.de (H.F.D.); ella.dennis@newcastle.ac.uk (E.P.D.); francesca.de-sousa-brito@newcastle.ac.uk (F.M.J.d.S.B.); louise.reynard@newcastle.ac.uk (L.N.R.); david.young@newcastle.ac.uk (D.A.Y.)

**Keywords:** skeletal dysplasia, protein misfolding, endoplasmic reticulum/cell stress, matrix metalloproteinase, biomarker

## Abstract

The intracellular retention of misfolded extracellular matrix proteins is a common disease mechanism in various rare skeletal diseases. This discovery has driven the study of ER stress and the unfolded protein response (UPR) as a promising therapeutic target in several skeletal dysplasias. In the case of *COL10A1* mutations, targeting the UPR resulted in a clinical trial of the repurposed drug carbamazepine; however, for other closely related skeletal disorders, treatment with carbamazepine was ineffective, indicating the need for suitable markers for in vitro screenings of potential drug treatments. Mutations in cartilage oligomeric matrix protein (COMP), a cartilage structural protein, cause both multiple epiphyseal dysplasia (MED) and pseudoachondroplasia (PSACH); together referred to as the COMPopathies, which result from the intracellular retention of mutant COMP to varying degrees. In contrast to other closely related skeletal disorders, caused by mutations in cartilage structural proteins, the involvement of the UPR is less clear, and so far, no common COMPopathy marker has been identified. Here, using cell models of COMPopathies, we identified *MMP9* upregulation as a common feature of six pathogenic COMP variants that do not induce a prominent UPR. We further show that the archetypal p.V194D matrilin-3 MED variant (which causes MED) does not induce *MMP9* expression, suggesting that MMP9 upregulation could serve as a specific marker of COMPopathies in vitro.

## 1. Introduction

Cartilage oligomeric matrix protein (COMP) is a secreted pentameric glycoprotein (524 kDa) predominantly present in cartilage, but also expressed in other connective tissues such as tendon and skin [1,2,3]. Functional studies have shown that COMP regulates collagen fibrillogenesis, contributes to ECM organisation, and can modulate growth factor signalling [4,5,6,7,8]. Structurally, COMP consists of an N-terminal coiled-coil domain, four epidermal growth factor (EGF)-like domains, eight calcium binding type 3 repeats (T3) and a C-terminal domain which acts as a site for collagen binding [5,9].

Mutations in COMP lead to two different, but clinically related, rare skeletal dysplasias: pseudoachondroplasia (PSACH) and multiple epiphyseal dysplasia (MED) [10,11]. The majority of *COMP* mutations are located in the T3 repeats, including the p.D469del mutation, which is found in approximately 30% of all PSACH cases [12], but is not seen in MED. Notably, PSACH is caused exclusively by mutations in *COMP*, whereas autosomal-dominant forms of MED can also be caused by mutations in matrilin-3 (*MATN3*) and type IX collagen (*COL9A1*, *COL9A2*, *COL9A3*) [13,14,15,16,17], other constituents of the cartilage ECM that have all been shown to interact directly with COMP [5,7,18].

In PSACH-MED, mutant COMP and mutant matrilin-3 are retained within the chondrocyte endoplasmic reticulum (ER) [19,20,21], triggering ER stress. Interestingly, previous studies identified three MED-causing COMP variants that have been described to exhibit incomplete mutant protein retention [22,23]. Finally, it has been demonstrated that induction of ER stress alone is sufficient to induce a chondrodysplasia-like phenotype in mice [24,25].

To restore ER homeostasis, in response to ER stress, and allow the survival of the cell, the unfolded protein response (UPR) drives a global reduction in protein translation and synthesis whilst enhancing the folding capacity of the ER by increasing chaperone protein and foldase levels (extensively reviewed in [26,27]). Despite initially acting as a pro-survival mechanism, persistent ER stress and continuous activation of all three branches of the UPR (controlled by IRE1, PERK and ATF6) can induce apoptosis. In contrast to mutations in *MATN3*, in which activation of the UPR has been unequivocally demonstrated in cell and mouse models [20,28,29], the role of the UPR for mutations in *COMP* is less clear [30,31,32]. Most studies to date have focused on the relatively common p.D469del COMP mutation and have proposed that inflammation and oxidative stress play a bigger role than the UPR [30,32,33].

Drug repurposing has emerged as a promising approach to treat skeletal diseases, which are often individually rare. For metaphyseal chondrodysplasia type Schmid (MCDS), caused by mutations in type X collagen, the identification of the UPR as a therapeutic target has driven the development of novel treatment strategies [34,35], and the repurposed drug carbamazepine (CBZ) has been tested in a clinical trial for children with MCDS [34]. Unfortunately, CBZ was unable to reduce the activation of the UPR following mutant matrilin-3 expression in cell models [36], highlighting the importance of appropriate model systems and drug screening strategies for distinct skeletal dysplasias. A recent study has shown that, with the use of an ER stress reporter in which an ER stress response element drives luciferase expression, curcumin was successful at reducing ER stress caused by mutant matrilin-3 accumulation by stimulating its degradation [37]. This discovery highlights that, in order to facilitate large-scale drug screenings in vitro, biomarkers are required that allow the efficient evaluation of compounds.

In this study, we aimed to understand the molecular mechanisms underlying COMPopathies using our in vitro model system in order to find a biomarker that could then be used to support drug repurposing/development. We demonstrate that *MMP9* expression is specifically upregulated in PSCAH and MED cell models of COMPopathies and that, in contrast, MED causing mutant matrilin-3 does not affect *MMP9* expression. MMP9 could therefore be used as a readout when screening for drugs to correct the molecular defect that occurs in COMPopathies.

## 2. Results

### 2.1. D469del COMP HT1080 Cells Are an In Vitro Model of COMPopathy

We have previously reported that GFP-tagged p.D469del COMP is retained intracellularly when expressed recombinantly in HT1080 human fibrosarcoma cells, in contrast to GFP-tagged wild-type (WT) COMP [32]. In agreement with our previous report, COMP was equally detected in the cell lysates of both wild-type (WT) and p.D469del GFP-tagged COMP-overexpressing cells (Figure 1A). When extracellular COMP levels were evaluated, WT COMP was readily detected in conditioned media, whereas p.D469del COMP was totally absent (Figure 1A). In the murine growth plate, the intracellular accumulation of p.D469del COMP disrupts chondrocyte proliferation and both increases and dysregulates apoptosis without triggering a conventional UPR [32]. In order to investigate whether the UPR is triggered when overexpressing pD469del COMP in human cells, markers of the three UPR branches were examined. When analysed by Western blotting, the levels of the chaperone BiP, the master regulator of the UPR, were slightly, but significantly, increased in the p.D469del COMP cell model (Figure 1B), whilst phosphorylation of the UPR PERK-effector eiF2α was not affected by p.D469del COMP (Figure 1C). Calnexin levels were also mildly elevated (Appendix A), whilst *XBP1* splicing, which is induced by IRE1/ERN1 during activation of the UPR, was reduced in p.D469del COMP cells (Appendix A). Despite no prominent activation of UPR signalling, levels of apoptosis were significantly increased (8.5 ± 0.2% vs. 4.0 ± 0.8%) in p.D469del COMP cells as demonstrated by TUNEL staining (Appendix A). Together, these findings replicate previous observations seen in mouse models of human COMPopathies [32,33] and thus demonstrate that cells overexpressing p.D469del COMP mimic the major aspects of PSACH pathology in vitro.

### 2.2. Transcriptomic Analysis of COMPopathy Cell Model

We aimed to employ our in vitro model system to identify potential biomarkers and/or gene signatures that are specific to COMPopathies and not present in skeletal dysplasias caused by other ECM proteins. Therefore, mRNA sequencing was performed on p.D469del and WT COMP cells. Out of the total of 12,487 genes expressed in these cells (transcripts per million (TPM) > 2), 465 genes were upregulated and 524 were downregulated, with a false discovery rate <0.05 and a fold change >1.5 (Figure 2A). KEGG pathway analysis was performed on all differentially expressed genes and revealed dysregulated cytokine signalling, including tumour necrosis factor (TNF) and IL-17 pathways, regulation of TRP channels by inflammatory mediators, and complement and coagulation cascades. ECM–receptor interaction, focal adhesion, organisation of actin cytoskeleton and cellular senescence were also amongst significantly enriched KEGG pathways (see Appendix A). To validate our mRNA sequencing, qRT-PCR was performed, including *MMP9* and *MMP1*, encoding matrix metalloproteinases (MMPs) that function to break down the cartilage matrix; *GALNT18*, which has recently been implicated in ER homeostasis [38]; and *SOX9*, encoding a transcription factor that is an established regulator of chondrogenesis [39] (Figure 2B). Interestingly, elevated MMP activity was previously demonstrated in an inducible mouse model of COMPopathies [40]. Furthermore, their expression and activity are known to be exacerbated in inflammatory signalling that has previously been suggested to play a role in the pathology of COMPopathies [41,42]. In agreement with the RNA sequencing findings, qRT-PCR confirmed that *MMP9, MMP1* and *GALNT18* were significantly upregulated in D469del COMP cells (9.4-, 3.3- and 18.7-fold, respectively); in contrast, *SOX9* expression was not changed (Figure 2B). These data therefore demonstrate that, similar to observations made in PSACH mouse models, p.D469del COMP cells display dysregulated cytokine signalling that could be responsible for a more inflammatory response than the classical ER stress-mediated UPR.

### 2.3. MMP9 Is Upregulated in Multiple COMPopathy Cell Models

The p.D469del mutation in COMP is responsible for approximately 30% of PSACH cases [12] and is therefore often used to study the molecular mechanisms underlying COMPopathies caused by mutations within the T3 repeats of COMP. Other mutations in these repeats, causing either PSACH or MED, remain less well characterised. We therefore analysed a selection of disease-causing COMP missense mutations to determine whether they had a similar effect on MMPs: the MED p.C312Y and p.D385N mutations and the PSACH mutations p.D473H (that is located within the same sequence of five aspartic acid residues as p.D469), p.G440R and p.D511Y [12,43]. When COMP levels were examined by Western blotting, the amount of extracellular COMP was significantly reduced by all the PSACH-causing mutations and the MED-causing p.D385N mutation, but not by p.C312Y (Figure 2C and S2A). Unlike the findings from the p.D469del COMP cell line, none of the selected mutations resulted in elevated levels of BiP or calnexin (Appendix A). We then investigated the effect of these mutations on *MMP9* expression using qRT-PCR (Figure 2D). Overexpression of all selected mutations resulted in an upregulation of *MMP9* expression. Strikingly, we observed not only a significant difference between wild-type and mutant COMP-overexpressing cells, but also a mutation-dependent increase in *MMP9* expression. The MED-causing COMP mutations p.C312Y and p.D385N led to 2.8- and 2.4-fold elevated *MMP9* expression, respectively, whilst for the PSACH-causing mutations p.D473H, p.G440R and p.D511Y, *MMP9* expression was elevated 5.7-, 4.3- and 20.4-fold, respectively (Figure 2D). This also resulted in enhanced MMP9 activity in conditioned media of p.C312Y, p.D385N and p.D473H and p.D511Y COMP cells, with a tendency for increased MMP9 activity in p.G440R COMP-expressing cells (Appendix A).

### 2.4. A Disease-Causing Matrilin-3 Mutation Does Not Cause Increased MMP9 Expression

Having demonstrated that *MMP9* expression is elevated in several in vitro models of COMPopathies, we next asked whether *MMP9* expression is altered in a model of another closely related skeletal dysplasia. To address this, we transiently overexpressed FLAG-tagged wild-type (WT) and mutant (p.V194D) matrilin-3 in HT1080 cells. The p.V194D variant of matrilin-3 is a typical MED-causing mutation and has been extensively characterised using both mouse and cell models [28,29,36]. In contrast to mutant COMP, p.V194D matrilin-3 has been demonstrated to activate a typical UPR in response to ER stress, both in vivo and in vitro.

When evaluated by Western blotting, WT matrilin-3 was present in cell lysates as well as conditioned media of transfected cells, whilst mutant p.V194D (VD) matrilin-3 was absent from conditioned media (Figure 3A). In untransfected cells (negative control, NC), no FLAG-tagged protein was observed in either the cell lysate or conditioned media (Figure 3A). Consistent with previous reports, expression of mutant p.V194D matrilin-3 resulted in a significant increase in BiP protein levels compared to WT matrilin-3 and untransfected cells (Figure 3A). *XBP1* splicing was also assessed to confirm activation of the IRE1 branch of the UPR (Figure 3B). In agreement with previous findings, *XBP1* splicing was significantly increased in mutant p.V194D compared to WT matrilin-3 (Figure 3B).

The increase in BiP protein levels was reflected in a significant upregulation of *HSPA5*, the gene that encodes BiP (Figure 3C). Thus, our model system appeared to reliably mimic the previously described features of *MATN3*-MED pathology. When we examined *MMP9* expression, we did not observe any changes (Figure 3C) compared to WT MATN3-Flag-expressing cells, therefore suggesting that *MMP9* is specifically elevated in response to mutant COMP but not mutant matrilin-3.

### 2.5. MMP9 Serum Levels in D469del COMP PSACH Mouse Model Are Not Altered

We next aimed to determine whether MMP9 levels were elevated in serum from a PSACH knock-in mouse model with the p.D469del COMP mutation [32]. However, there were no significant sex differences in MMP9 serum levels from either genotype (Appendix A), and in contrast to our findings in vitro, we did not detect a significant increase in MMP9 serum levels from either female or male mice (Figure 4A,B). Indeed, when data from both sexes were combined, MMP9 levels were significantly decreased (mean ng/mL in WT and Y ng/mL in D469del) in the D469del COMP PSACH mouse model (Figure 4C).

## 3. Discussion

Skeletal dysplasias are individually rare diseases; however, together, they affect approximately 1 in 4000 people. The characterisation of disease mechanisms and the identification of ER stress as a crucial component have driven the development of novel treatment approaches. Nevertheless, to perform large-scale drug screenings, suitable (bio)markers need to be available.

In contrast to mutations in *COL10A1* and *MATN3*, the UPR pathways appear to be largely unaffected by the retention of mutant COMP in the ER in animal models of MED and PSACH-causing COMP mutations [27,32]. Herein, we demonstrate the absence of pronounced activation of the UPR in human cells overexpressing the archetypal p.D469del COMP mutation as well as for several missense mutations, which renders components of the UPR unsuitable as biomarkers for COMPopathies.

The specific reason why mutant matrilin-3 and type X collagen but not COMP trigger the UPR remains to be elucidated in future studies. Similar findings have been reported for mutations in *COL1A1* and *COL1A2* that cause osteogenesis imperfecta, and whilst some mutations trigger the UPR, others do not [44,45]. It is tempting to speculate that, for collagens, the location of the mutation is crucial, and that mutations in the pro-peptide of type I collagen trigger increased binding of chaperones such as BiP, whilst mutations in the triple-helical region do not. Why mutant COMP would not be recognised by chaperones in the same way as matrilin-3 or various collagens remains to be investigated.

Our study used transcriptomic analysis and multiple mutant COMP variants to discover potential biomarkers of COMPopathies. Our reported upregulation of *MMP9* in COMPopathy cell models complements previous observations from mouse models, which have described a more inflammation-mediated disease mechanism and generally elevated MMP activity [32,33,40]. In agreement with this, MMP9 expression has been described to be stimulated in chondrocytes and synoviocytes by the inflammatory cytokines IL-1β and TNFα [46]. However, we found MMP9 serum levels in D469del COMP PSACH mice to be reduced compared to those in WT mice rather than elevated. This discrepancy could be caused by several factors. MMP9 levels may be increased locally, but this may not be detectable in serum. Since some cartilage matrix molecules have a long half-life and most cartilage is actively built and remodelled during the growth period, it is also possible that the physiological expression levels of p.D469del COMP do not stimulate MMP9 expression in young adult mice. Additionally, other mechanisms that control MMP9 levels may interfere with or even counteract the stimulation of MMP9 expression by mutant COMP accumulation in vivo.

In the absence of a complete understanding of the disease mechanism, the ability to perform drug screenings of large compound libraries remains ever more important. Our work demonstrates that several MED/PSACH-causing COMP mutations result in increased *MMP9* expression. Increased *MMP9* expression was even observed in cells expressing the p.C312Y COMP mutation, which did not affect COMP secretion significantly. Importantly, *MMP9* expression was unaffected by the expression of MED causing p.V194D matrilin-3 mutation, suggesting it is specifically differentially regulated in cell models of COMPopathies. Since MMP9 is known to be involved in inflammatory processes, it is our hope that our findings will lead to the discovery of compounds that can reduce inflammation and improve chondrocyte pathology in COMPopathies. Since PSACH and MED are rare diseases, and resources are often limited, our work suggests that *MMP9* expression could act as a suitable RNA biomarker for compound screenings independently of the specific *COMP* mutation.

## 4. Materials and Methods

### 4.1. Generation and Culture of Wild-Type and Mutant COMP Cell Lines

FACS-sorted pEGFP-N3 hCOMP wild-type (WT) and p.D469del COMP HT1080 cells have been described previously [32]. HT1080 cells were obtained from ATCC (ATCC, Manassas, VA, USA; ATCC CCL0121). All other constructs (p.C312Y, p.D385N, p.D473H, p.G440R and p.D511Y) were generated using the pEGFP-N3 hCOMP construct as a template; primers containing the desired mutations (see Appendix A were made and mutagenesis was performed using the QuikChange Site-directed mutagenesis kit (Agilent Technologies UK Ltd., Edinburgh, UK) according to the manufacturer’s instructions. HT1080 cells were maintained, transfected and selected as described previously [32].

### 4.2. SDS-PAGE and Western Blotting

Cell lysates and conditioned media were collected 72 h after confluency. Conditioned medium was collected and briefly centrifuged to remove debris before 5XSDS loading buffer (625 mM Tris-base, 50% (*v*/*v*) Glycerol, 10% (*w*/*v*) SDS, 0.025 (*w*/*v*) Bromophenol blue pH 6.8) was added. Cell layers were washed with sterile PBS and scraped in 2 × SDS loading buffer containing protease inhibitors (F. Hoffmann-La Roche AG, Basel, Switzerland). Lysates were passed through an insulin syringe and centrifuged for 10 min at 13,000× *g* at 4 degrees. DTT was added to the lysates and conditioned media before the samples were boiled. The samples were loaded onto pre-cast Novex NuPAGE 4–12% Bis-Tris gels (Thermo Fisher Scientific Inc., Waltham, MA, USA) and transferred onto a nitrocellulose membrane. The membrane was stained with Ponceau staining solution for 2 min and, after imaging, blocked in 3% BSA/TBS-T for 1 h at room temperature (RT). Incubation with primary antibody was carried out overnight at 4 °C in blocking solution using the appropriate dilution (see Appendix A). Incubation with secondary antibody was carried out for 1 h at RT in 5% skimmed milk/TBS-T. Proteins of interest were detected using SuperSignal West Pico Plus Chemiluminescent Substrate (Thermo Fisher Scientific Inc.). Band intensities were quantified in ImageJ 1.54F (National Institutes of Health, Bethesda, MD, USA) using GAPDH a s loading control for standard normalisation of band intensities across the different biological and technical experimental replicates.

### 4.3. TUNEL Assay

Apoptosis was analysed using a DeadEnd fluorometric TUNEL assay (Promega, Southampton, UK) according to the manufacturer’s instructions. Nuclei were stained with Fluoroshield mounting medium with DAPI (Abcam, Cambridge, UK).

### 4.4. RNA Sequencing

RNA was isolated from p.D469del and WT COMP cells using the EZNA DNA/RNA extraction kit (Omega Bio-tek, Inc, Norcross, GA, USA) according to the manufacturer’s instructions, before purified RNA was treated with a DNA-free kit (Thermo Fisher Scientific Inc.) to remove DNA contamination.

RNA sequencing was performed by the Genomic Core Facility at Newcastle University using the NextSeq500 platform. Transcriptome analysis was then performed as described previously [47] using hg38 as a reference, and with genes exhibiting a log_2_ fold change larger than ±0.58 (=1.5-fold change) and adjusted *p*-value smaller than 0.05 considered significantly differentially expressed. KEGG pathway analysis was carried out using the Enrichr tool [48].

### 4.5. Gene Expression Analysis by qRT-PCR

RNA was extracted from cells using the ReliaPrep RNA Mini prep kit (Promega) according to the manufacturer’s instructions. A 1 µg amount of RNA was transcribed into cDNA using the GoScript RT kit (Promega) according to the manufacturer’s instructions. After cDNA synthesis, samples were treated with 1 U of RNaseH (Thermo Fisher Scientific Inc.) for 20 min at 37 °C. Gene expression was measured using the PowerUp SYBR green master mix (Thermo Fisher Scientific Inc.) according to the manufacturer’s instructions using the primers specified below (see Appendix A).

For *XBP1* splicing analysis, BioMix Red master mix (Bioline Reagents Ltd., London, UK) was used according to the manufacturer’s instructions. The long and short forms of the *XBP1* transcript were visualised using agarose gel electrophoresis.

### 4.6. In-Gel Zymography

A 1 mL volume of serum-free cell culture supernatant was concentrated to 25 µL using VivaSpin columns (Sigma Aldrich Ltd., Haverhill, UK), with a molecular weight cut-off of 10 kDa, for 10 min at 12,000× *g*. The concentrated sample was then mixed with the appropriate amount of 5 × SDS loading buffer (without DTT) and incubated for 10 min at RT. Non-reducing SDS-PAGE was performed on ice using 7.5% SDS polyacrylamide gels containing 0.3% gelatine. After washing in 0.25% Triton-X100, the zymogram was incubated overnight at 37 °C and 60 rpm in digestion buffer (50 mM Tris-HCl pH 7.4, 10 mM CaCl_2_, 0.2% NaN_3_). The reaction was stopped by staining the zymogram in Coomassie staining solution (40% methanol, 10% acetic acid, 0.2% Coomassie Brilliant Blue G250) for 60 min. An identical gel without gelatine was run in parallel, stained with Coomassie solution, with its band intensities used as loading controls.

### 4.7. Serum MMP9 ELISA

Blood samples were extracted from 8-week-old unfastened wild-type and p.D469del mutant COMP mice (described in detail in [32]) by terminal cardiac puncture under anaesthesia. Briefly, blood was drawn from the left ventricle using a 25G needle and 1 mL syringe and transferred directly into a Microtainer^TM^ serum separator tube (BD, Franklin Lakes, NJ, USA) containing anti-coagulant. Whole blood was allowed to clot for 30 min at room temperature before being centrifuged at 1500× *g* and 4 °C for 20 min. The level of MMP9 in the resulting serum was then deduced using the R&D Systems Mouse Total MMP9 Quantikine ELISA kit (Biotechne Ltd., Abingdon, UK) according to the manufacturer’s instructions. All the in vivo work was carried out under project licence PPL60/4525 in compliance with the Animals Scientific Procedures Act (ASPA) 1986 according to Directive 2010/63/EU of the European Parliament.

### 4.8. Statistical Analysis

Statistical analysis was performed in R studio using either unpaired two-tailed Student’s t-tests or two-way ANOVA and Tukey post hoc tests (for multiple groups). For gene expression analysis, fold changes were first converted into log_2_ fold changes prior to statistical analysis. A *p*-value < 0.05 was considered significant.

## Figures and Tables

**Figure 1 ijms-26-12070-f001:**
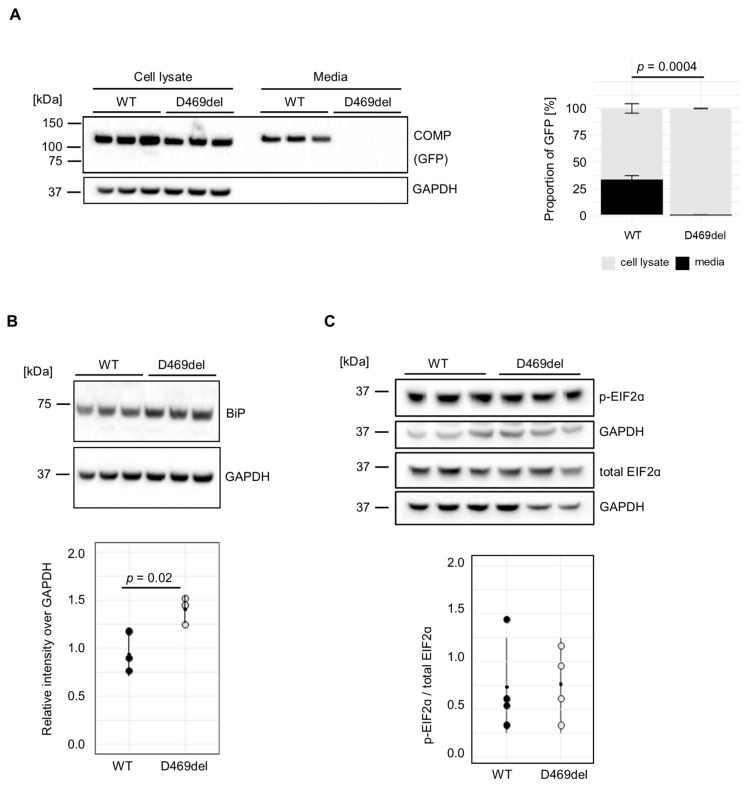
Overexpression of p.D469del COMP does not induce a classical unfolded protein response. (**A**) Representative Western blot and quantification (3 experiments) of cell lysates and conditioned media of wild-type (WT) and p.D469del COMP-overexpressing HT1080 cells 72 h after confluency. COMPs protein was detected via the C-terminal GFP-tag. GAPDH was used as a loading control. (**B**) Representative Western blot and quantification of BiP protein levels in wild-type (WT) and p.D469del COMP-overexpressing HT1080 cells 72 h after confluency. GAPDH was used as a loading control. Error bars represent standard deviations between 3 experiments. BiP was detected using the same membrane as in (**A**); therefore, the same GAPDH loading control is shown. (**C**) Representative Western blot of cell lysates from WT and p.D469del COMP-overexpressing HT1080 cells using a phospho-eiF2α and a total eiF2α-specific antibody. GAPDH was used as a loading control. Means and standard deviations of 4 experiments are shown. (**A**–**C**) Student’s *t*-test was used to determine *p*-values.

**Figure 2 ijms-26-12070-f002:**
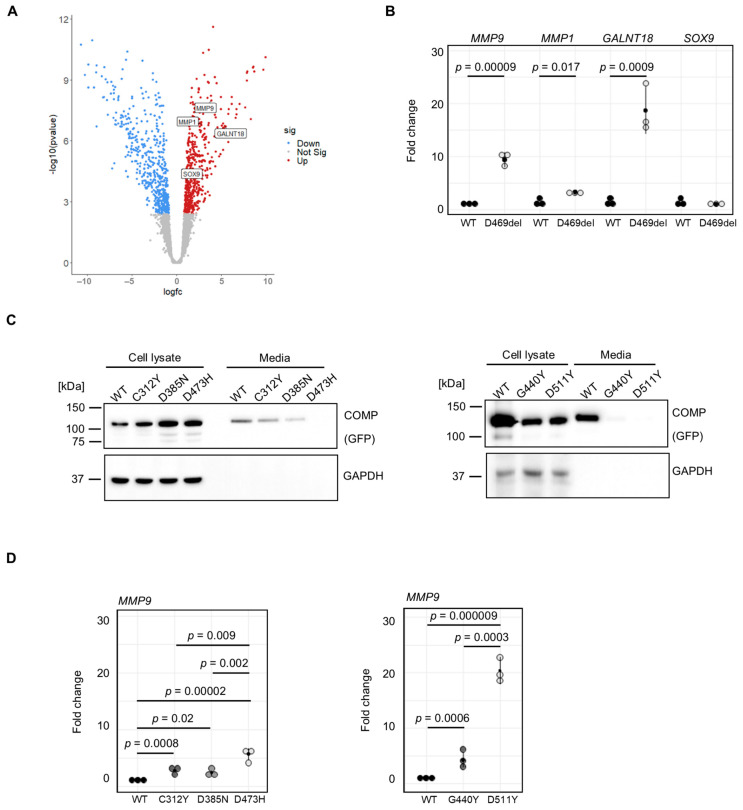
MMP9 is upregulated in cell models of COMPopathies. (**A**) Volcano plot of differentially expressed genes. (**B**) Validation of differential expression (log_2_ fold change) of *MMP9*, *MMP1*, *GALNT18* and *SOX9* by qRT-PCR. Means and standard deviations of 3 replicates are shown. Student’s *t*-test was used to determine *p*-values. (**C**) Representative Western blot of cell lysates and conditioned media of wild-type (WT) and CY (p.C312Y, MED), DN (p.D385N, MED), DH (p.D473H, PSACH), GR (p.G440R, PSACH) and DY (p.D511Y, PSACH)-overexpressing HT1080 cells 72 h after confluency. COMPs were detected via the C-terminal GFP-tag. GAPDH was used as a loading control. (**D**) *MMP9* expression was assessed by qRT-PCR 72 h after confluency. *18S* was used as a housekeeping gene. Means and standard deviations of 3 replicates are shown; *p*-values were determined using ANOVA and Tukey post hoc tests.

**Figure 3 ijms-26-12070-f003:**
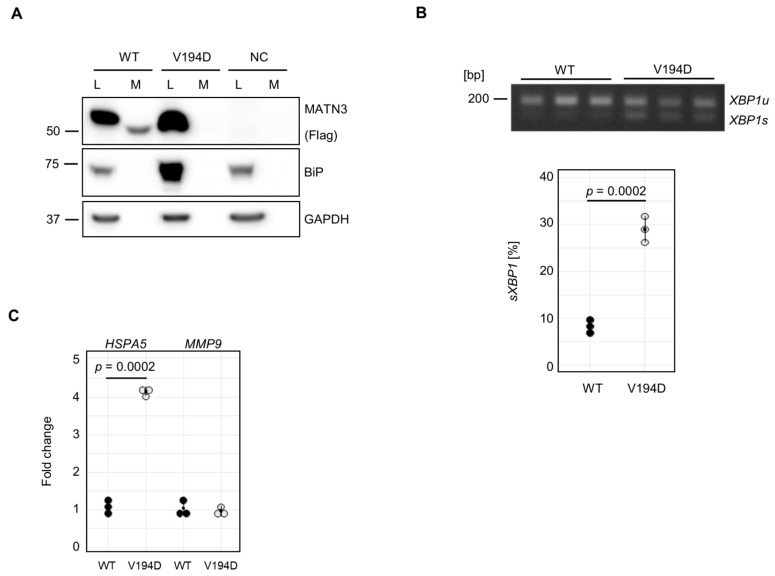
Expression of mutant matrilin-3 (MED) does not drive *MMP9* expression. (**A**) Representative Western blot of flag-tagged matrilin-3 and BiP protein levels in untransfected (NC) WT or VD transfected cells 48 h after transfection. GAPDH was used as loading control. Means and standard deviations of 3 experiments are shown. (**B**) *XBP1* splicing was analysed in wild-type and mutant VD matrilin-3 overexpressing HT1080 cells using RT-PCR and agarose gel electrophoresis. Means and standard deviations of 3 replicates are shown. (**C**) Analysis of *HSPA5* and *MMP9* expression by qRT-PCR. *18S* was used as a housekeeping gene. Means and standard deviations of 3 replicates are shown. (**B**,**C**) Student’s *t*-test was used to determine *p*-values.

**Figure 4 ijms-26-12070-f004:**
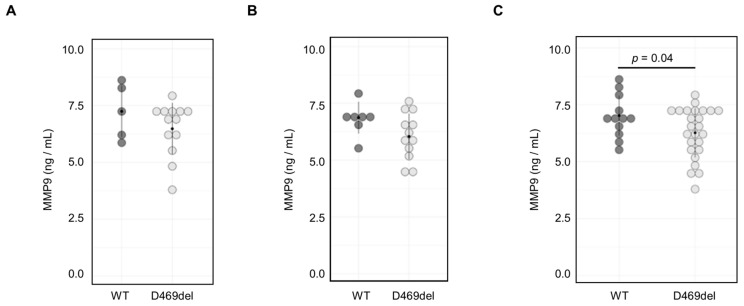
MMP9 serum levels are not upregulated in the PSACH mouse model. Serum samples from (**A**) female wild-type (N = 5) and D469del COMP PSACH mice (N = 13) and (**B**) male wild-type (N = 7) or p.D469del COMP PSACH mice (N = 12) were analysed by ELISA. (**C**) MMP9 serum levels in wild-type (N = 12) and D469del COMP PSACH mice (N = 25) of both sexes. Student’s *t*-test was applied to determine *p*-values.

## Data Availability

Additional data are available from the corresponding author upon reasonable request.

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
