# Peer review of "Elevated MMP9 Expression—A Potential In Vitro Biomarker for COMPopathies"

_ijms, 2025, doi:10.3390/ijms262412070_

Round 1
Reviewer 1 Report
Comments and Suggestions for Authors
This paper describes the in vitro effects of COMP mutations on MMP9 expression and suggests its potential use as a biomarker. The manuscript is well-organized and written, with appropriate figures and tables, describing the in vitro results. The most interesting outcome is the differences in MMP9 activation by different COMP mutations versus a matrilin-3 mutation. The use as a biomarker is hardly proven by these studies.
- Do these results translate in their PSACH mouse model growth plate chondrocytes/joints? This is important as MMP9 from their mouse model blood samples was not increased.
- Why did the authors focus on MMP9 when review of the literature shows that another PSACH mouse model showed tissue elevation of MMP13 and 2? MMP13 is more chondrocyte-specific then MMP9, even though MMP9 is part of the inflammatory response. The authors mention the other study showing MMP13 increased expression but not why they didn’t confirm that result.
- Discussion section – Second paragraph – ref is missing.
- Supplemental Table 1 in section 2.2 refers to up- and down-regulated genes but in the Supplementary section it is a list of primers.
Author Response
This paper describes the in vitro effects of COMP mutations on MMP9 expression and suggests its potential use as a biomarker. The manuscript is well-organized and written, with appropriate figures and tables, describing the in vitro results. The most interesting outcome is the differences in MMP9 activation by different COMP mutations versus a matrilin-3 mutation. The use as a biomarker is hardly proven by these studies.
We agree that the discrepancy between the in vitro and in vivo studies was unexpected; however, as stated in the manuscript our aims were to identify potential in vitro markers to aid in drug discovery screening.
- Do these results translate in their PSACH mouse model growth plate chondrocytes/joints? This is important as MMP9 from their mouse model blood samples was not increased.
We no longer have access to growth plate tissue from these mice, but as stated above our intention was to identify markers to use with cell lines.
- Why did the authors focus on MMP9 when review of the literature shows that another PSACH mouse model showed tissue elevation of MMP13 and 2? MMP13 is more chondrocyte-specific then MMP9, even though MMP9 is part of the inflammatory response. The authors mention the other study showing MMP13 increased expression but not why they didn’t confirm that result.
As stated in the manuscript MMP9 was identified as elevated by RNAseq. Our mouse model is very different to the one referenced (i.e. targeted knock in vs inducible transgenic overexpression of mutant human COMP) and therefore we did not see the value of confirming those findings.
- Discussion section – Second paragraph – ref is missing.
Corrected
- Supplemental Table 1 in section 2.2 refers to up- and down-regulated genes but in the Supplementary section it is a list of primers.
We apologise for this discrepancy. Suppl. Table 1 was inadvertently from an early draft of the manuscript. We subsequently decided the present the RNAseq data as a Volcano Plot (Fig 2A) and as KEGG analysis (Table 1). We have therefore removed any reference to this Table. In addition on review of section 2.2 it became apparent that there was some Figure numbers and text discrepancies – this has now been rectified.
Reviewer 2 Report
Comments and Suggestions for Authors
- Abstract: Line 11: The intracellular retention of misfolded extracellular matrix proteins is a common disease mechanism in various individually rare skeletal diseases. The meaning of individually rare skeletal diseases is unclear.
- Mutations in cartilage oligomeric matrix protein (COMP), a cartilage structural protein, causes both multiple epiphyseal dysplasia (MED) and pseudoachondroplasia (PSACH); together referred to as COMPopathies, and result in intracellular retention of mutant COMP protein to varying degrees: Please consider removing 'and' highlighted here and add results in.
- Fig1A B and C, immunoblots containing the housekeeping/loading control GAPDH shows unequal loading, therefore the results and conclusion noted are not interpretable
- Please check the immunoblot results in all the figures and conclude accordingly.
Author Response
- Abstract: Line 11: The intracellular retention of misfolded extracellular matrix proteins is a common disease mechanism in various individually rare skeletal diseases. The meaning of individually rare skeletal diseases is unclear.
We have removed ‘individually’.
- Mutations in cartilage oligomeric matrix protein (COMP), a cartilage structural protein, causes both multiple epiphyseal dysplasia (MED) and pseudoachondroplasia (PSACH); together referred to as COMPopathies, andresult in intracellular retention of mutant COMP protein to varying degrees: Please consider removing 'and' highlighted here and add results in.
Replaced with:-
Mutations in cartilage oligomeric matrix protein (COMP), a cartilage structural protein, cause both multiple epiphyseal dysplasia (MED) and pseudoachondroplasia (PSACH); together referred to as the COMPopathies and which result from the intracellular retention of mutant COMP protein to varying degrees.
- Fig1A B and C, immunoblots containing the housekeeping/loading control GAPDH shows unequal loading, therefore the results and conclusion noted are not interpretable
- Please check the immunoblot results in all the figures and conclude accordingly.
GAPDH was used to normalise band intensities as per standard experimental procedures. This has now been clarified in the methods section. Each experiment was repeated to provide biological replicates of n=3. This analysis in also described in Supplemental Figure 2. We have revised this sentence for further clarification:
“Band intensities were quantified in ImageJ/Fiji (National Institutes of Health, Bethesda, MD, USA) using GAPDH as loading control for standard normalization of band intensities across the different biological and technical experimental replicates.”
Round 2
Reviewer 2 Report
Comments and Suggestions for Authors
All comments were sufficiently resolved